# Diversity and plant growth-promoting functions of diazotrophic/N-scavenging bacteria isolated from the soils and rhizospheres of two species of *Solanum*

**Mónica Yorlady Alzate Zuluaga**[1], **Karina Maria Lima Milani**[1], **Leandro Simões Azeredo Gonçalves**[2], **André Luiz Martinez de Oliveira**[1]*

1 Departamento de Bioquímica e Biotecnologia, Universidade Estadual de Londrina, Londrina, Paraná, Brazil, 2 Departamento de Agronomia, Universidade Estadual de Londrina, Londrina, Paraná, Brazil

* almoliva@uel.br

**Data Availability Statement:** All relevant data are within the paper and its Supporting Information files. All nucleotide sequence files are available

## Abstract

Studies of the interactions between plants and their microbiome have been conducted worldwide in the search for growth-promoting representative strains for use as biological inputs for agriculture, aiming to achieve more sustainable agriculture practices. With a focus on the isolation of plant growth-promoting (PGP) bacteria with ability to alleviate N stress, representative strains that were found at population densities greater than $10^4$ cells $g^{-1}$ and that could grow in N-free semisolid media were isolated from soils under different management conditions and from the roots of tomato (*Solanum lycopersicum*) and lulo (*Solanum quitoense*) plants that were grown in those soils. A total of 101 bacterial strains were obtained, after which they were phylogenetically categorized and characterized for their basic PGP mechanisms. All strains belonged to the Proteobacteria phylum in the classes Alphaproteobacteria (61% of isolates), Betaproteobacteria (19% of isolates) and Gammaproteobacteria (20% of isolates), with distribution encompassing nine genera, with the predominant genus being *Rhizobium* (58.4% of isolates). Strains isolated from conventional horticulture (CH) soil composed three bacterial genera, suggesting a lower diversity for the diazotrophs/N scavenger bacterial community than that observed for soils under organic management (ORG) or secondary forest coverture (SF). Conversely, diazotrophs/N scavenger strains from tomato plants grown in CH soil comprised a higher number of bacterial genera than did strains isolated from tomato plants grown in ORG or SF soils. Furthermore, strains isolated from tomato were phylogenetically more diverse than those from lulo. BOX-PCR fingerprinting of all strains revealed a high genetic diversity for several clonal representatives (four *Rhizobium* species and one *Pseudomonas* species). Considering the potential PGP mechanisms, 49 strains (48.5% of the total) produced IAA (2.96–193.97 µg IAA mg protein$^{-1}$), 72 strains (71.3%) solubilized FePO$_4$ (0.40–56.00 mg l$^{-1}$), 44 strains (43.5%) solubilized AlPO$_4$ (0.62–17.05 mg l$^{-1}$), and 44 strains produced siderophores (1.06–3.23). Further, 91 isolates (90.1% of total) showed at least one PGP trait, and 68 isolates (67.3%) showed multiple PGP traits. Greenhouse trials using the bacterial collection to inoculate tomato or lulo plants revealed increases in plant biomass (roots, shoots or both plant

from the NCBI database (GenBank accession number KX884880 to KX884993).

**Funding:** This work was supported by the INCT - Plant Growth-Promoting Microorganisms for Agricultural Sustainability and Environmental Responsability (INCT-MPCPAgro), Fundação Araucária (conv. no. 309/2012), CNPq (conv. No. 458099/2014-7) and CAPES. The funders had no role in study design, data collection and analysis, decision to publish, or preparation of the manuscript.

**Competing interests:** The authors have declared that no competing interests exist.

tissues) elicited by 65 strains (54.5% of the bacterial collection), of which 36 were obtained from the tomato rhizosphere, 15 were obtained from the lulo rhizosphere, and 14 originated from samples of soil that lacked plants. In addition, 18 strains showed positive inoculation effects on both *Solanum* species, of which 12 were classified as *Rhizobium* spp. by partial 16S rRNA gene sequencing. Overall, the strategy adopted allowed us to identify the variability in the composition of culturable diazotroph/N-scavenger representatives from soils under different management conditions by using two *Solanum* species as trap plants. The present results suggest the ability of tomato and lulo plants to enrich their belowground microbiomes with rhizobia representatives and the potential of selected rhizobial strains to promote the growth of *Solanum* crops under limiting N supply.

## Introduction

The development of sustainable and ecological methods to improve agricultural productivity without expanding the cultivable land area has become a crucial challenge to ensure food security. Among the different tools available to increase productivity in agriculture, the use of plant growth-promoting bacteria (PGPB) has demonstrated great potential to achieve the goal of sustainability. PGPB can directly contribute to plant nutrition and can alleviate the detrimental effects caused by biotic and abiotic stresses to which field crops are subjected; PGPB can also indirectly improve soil fertility and quality by at least partially substituting for the use of chemical inputs [1–4]. Nevertheless, despite the large number of studies reporting the advantages of the use of PGPB as inoculants within commercial agricultural crops, their use as a regular agricultural practice on nonleguminous crop species is still underexplored in comparison with the amount of agricultural lands annually cultivated worldwide [5]. Major constraints for the broad use of plant growth-promoting bacterial inoculants are due to poor knowledge about the variability in the structure, composition and function of the plant microbiome subjected to the influence of plant genotypes on different soil microbial communities and edaphoclimatic conditions. Therefore, studies focused on the ecology, physiology and biochemistry of plant-microbe interactions are needed to deepen the understanding of the microbiome role in plant fitness and to facilitate the development of highly effective biological inputs for agriculture. In this sense, the application of culture-dependent methods to study plant-bacterium interactions can provide resources to elucidate the ecological and physiological role of plant microbiome representatives *in vitro* and determine their effects when present within host plant [6].

The rhizosphere, which is defined as a thin soil layer close to the root system and is actively influenced by metabolic activity [7], has often been used as the preferential site for the isolation of PGPB with potential applications as biofertilizers [8,9]. The habitat of rhizospheric soils differs physically, chemically and biologically from the nonrhizospheric soils (or bulk soil) because of the continuous release of a complex array of organic-based molecules such as rhizodeposits [10,11]. Qualitative and quantitative variations in rhizodeposits are thought to play an active role in shaping the rhizosphere microbiome by enriching population sizes of microbial representatives with competence to consume released compounds according to the plant genotype, age and local environmental conditions [12,13]. The modulation of the microbial community of plant holobionts is not stable over time because different biotic and abiotic conditions normally occur throughout the plant life cycle, and these variations may produce greater impacts than the genomic content on the qualitative and quantitative composition of rhizodeposits [14,15]. Other biotic and abiotic factors, such as the native soil microbial

diversity, physical and chemical soil properties and geographic and climatic conditions, also play important roles in shaping the composition and function of rhizosphere microbiomes [16,17].

In general, the bacterial diversity found in the rhizosphere is represented mainly by species belonging to the Proteobacteria, Firmicutes and Actinobacteria phyla, where the most common genera reported include *Bacillus*, *Pseudomonas*, *Enterobacter*, *Arthrobacter*, *Rhizobium*, *Agrobacterium*, *Burkholderia*, *Azospirillum*, *Azotobacter*, *Mycobacterium*, *Flavobacterium*, *Cellulomonas* and *Micrococcus* [18–20]. Several bacteria from these phylogenetic clades are reported to have the ability to benefit plant growth and hence are considered PGPB, although the establishment of functional relationships between plants and representatives from the soil microbiome indicates coevolutionary requirements [10,21,22]. The influence of a given plant growth-promoting bacterial strain on plant development and growth succeeds via complex chemical crosstalk responsible for eliciting, in general, multiple PGP effects that involve direct and/or indirect growth-promoting mechanisms. Direct growth-promoting mechanisms involve the facilitation of nutrient acquisition or modulation of development by interfering in the balance of endogenous plant hormones, while indirect growth-promoting mechanisms are related to biocontrol, increased resistance against stresses and the modulation of plant gene expression [23–25].

The development of plant growth-promoting bacterial bioinputs ultimately requires the collection of rhizocompetent strains and the elucidation of their role in plant growth and health, their resilience as a holobiont component and their ecological relationships within the plant rhizosphere, where important plant-plant growth-promoting bacterial associations occur. In this sense, the present study describes the isolation, identification and characterization of culturable bacteria that are diazotrophic or have an N-scavenging ability and that were obtained from soils under different management conditions or from the roots plus the rhizosphere soil of tomato and lulo plants grown in these same soils. The PGP mechanisms of the isolated strains were characterized, aiming to identify those with potential for use as bioinputs targeted to alleviate N stress. Moreover, this study also aimed to address whether changes in the diversity of the target bacterial groups studied were related to soil management practices and/or the plant species used as traps for bacterial isolation.

## Materials and methods

### Soil characterization and sampling

Soils under three different management conditions were collected from the 0 to 10 cm depth from the experimental farm of Universidade Estadual de Londrina (Londrina, Parana, Brazil; 23˚20'31"S, 51˚12'38"W), aiming to assess microbial communities with distinct structural compositions. The management conditions included a secondary forest with no agricultural use for at least 35 years (SF), a site under intensive horticultural cropping with conventional management practices (CH) and a site under intensive horticultural cropping with organic management (ORG) practices. The conventional management site received macronutrients (NPK, 15-15-20, 100 g m$^{-2}$) and micronutrients (FTE BR-12, 20 g m$^{-2}$) every three months, while the organic site was regularly supplemented with organic compost produced from cow dung (2 kg m$^2$). All soil samples were collected at geographically close sites (~400–600 m distance) and were classified as Oxisols (Latossolo Vermelho Eutroférrico, Brazilian classification), with a high clay content (78%). A chemical analysis of the soil samples is presented in Supporting Information S1 Table.

## Isolation of diazotrophic/N-scavenging bacteria

The experimental units used to assess the associated diazotrophic or N-scavenging bacteria were pots filled with 2 kg of soil from the different management conditions (SF, CH or ORG). The pots were sown with three surface-disinfested seeds [26] of tomato (*Solanum lycopersicum* cv. Santa Cruz Kada Gigante, Topseed, Santo Antonio da Posse, Brazil) or lulo (*Solanum quitoense* Lam., kindly provided by the Experimental Station Vila Anamaria, Londrina, Brazil). In total, six different conditions were established to obtain diazotrophic/N scavenger isolates: combinations of the three soil conditions and the two *Solanum* species, with three replicates each. The plants were grown under greenhouse conditions in a random experimental design, and at 15 days after sowing, the seedlings were thinned to one healthy plant per pot for an additional growth period of 30 (tomato) or 60 (lulo) days, with tap water used to water the pots every two days. After the growth period, the plants were gently removed from the pots and shaken to remove loosely adhered soils from their roots. Unwashed roots (roots with tightly adhered rhizospheric soil) from both tomato and lulo plants (TR and LR treatments) were collected from each replicate, and 1 g samples were taken from the middle third of root systems and used to obtain homogenates in sterile saline solution (0.85%), following serial dilutions (up to $10^{-6}$) of sterile saline solutions. Samples from soils under the different management conditions were also subjected to isolation of diazotrophic/N-scavenging bacteria concomitantly with the establishment of the plant growth experiment to obtain a picture of the diversity and population sizes of these microbial groups in the absence of plants. To this end, 10 g samples from each soil condition were transferred to sterile Petri dishes, to which 2 ml of sterile distilled water was added; afterward, the dishes were incubated at 28°C for 48 h to restore microbial activity, using three replicates for each soil condition. After incubation, the samples were transferred to Erlenmeyer flasks (250 ml volume) supplemented with 90 ml of sterile saline solution, incubated under orbital shaking (180 rpm for 30 min) and then serially diluted (up to $10^{-6}$) in sterile saline solutions.

Isolation of diazotrophic/N-scavenging bacteria was performed according to the methods of Baldani et al. [27], with minor modifications. Briefly, aliquots of 0.1 ml of relatively strong dilutions ($10^{-4}$ to $10^{-6}$) of all samples were inoculated into vials containing 5 ml of semisolid N-free media (JMV, JNFb, NFb, LGI or LGI-P) and incubated for seven days at 28°C. After the incubation period, the vials with a veil-like pellicle near the surface of the culture medium were considered positive for the presence of diazotrophic/N-scavenging bacteria. A loopful was taken from the pellicle of each culture medium and used to inoculate fresh N-free semi-solid media of the same type; this step was repeated three more times to assure the ability of isolates to grow in the N-free culture media. After confirmation of growth in the N-depleted culture media, the population densities of diazotrophic/N-scavenging bacteria were determined by estimating the most probable number (MPN) following purification of the strains from the strongest dilutions, after which they were stored at -20°C for further analysis.

## DNA extraction and 16S rRNA gene sequencing

Genomic DNA for each bacterial isolate was extracted according to the methods of Cheng and Jiang [28] using bacterial biomass obtained after growth in DYGS liquid media [25] for 48 h and incubation at 28°C under 180 rpm on an orbital shaker Tecnal TE-424 (Piracicaba, Brazil). Partial 16S rRNA gene amplification was performed using the primers Y1 and Y3 according to the methods of Koskey et al. [29]. The resulting amplicons were purified with the aid of an ExoSAP-IT kit (USB Corp., USA) according to the manufacturer's instructions. The amplicons were sequenced using 362F primers [30] in conjunction with a BigDye Terminator v3.1 Cycle Sequencing Kit and an ABI Prism Genetic Analyzer 3100 (Applied Biosystems, Foster City,

USA), according to the manufacturer's instructions. The 16S rRNA sequences of the isolates were trimmed for quality using BioNumerics v. 4.6 software (Applied Maths, Kortrijk, Belgium) and deposited in the GenBank database under accession numbers KX884880–KX884993.

### Sequence analysis and BOX-PCR fingerprinting

Before performing the phylogenetic analysis of the 16S rRNA gene sequences, they were subjected to the Ribosomal Database Project II (RDP) classifier [31] for phylogenetic positioning at the genus level (95% confidence threshold). Phylogenetic relationships between each isolate strain and the species of the respective genera were reconstructed using MEGA 7.0 software after alignment with the 16S rRNA gene sequences of type strains retrieved from the RDP database [32]. Phylogenetic trees were generated by the neighbor-joining (NJ) method with the Kimura 2 parameter (K2P). Nodal robustness of the trees was assessed using 1000 bootstrap replicates, and sequence identity was estimated using the resulting sequence identity matrix.

To evaluate the genomic diversity of the isolated strains, the fingerprinting of the genomic DNA for each strain was obtained via BOX-PCR using BOX A1R primers [33]. The PCR cycling and gel electrophoresis were performed according to the methods of Rademaker and de Bruijn [34]. The resulting fingerprints were analyzed using BioNumerics v. 4.6 to produce a dendrogram based on the Jaccard coefficient from the distance matrix (2% tolerance in terms of band size) and the unweighted pair-group method with arithmetic mean (UPGMA). A 70% cutoff was chosen to define the similarity clusters, as suggested in previous diversity studies related to rhizosphere-associated bacteria [35,36].

### *In vitro* characterization of plant growth-promoting traits

All isolates with a putative role in plant growth-promotion were evaluated for the following traits: synthesis of indole-3-acetic acid (IAA), production of siderophores and solubilization of two insoluble phosphate sources ($FePO_4$ and $AlPO_4$). These assays were carried out using bacterial suspensions grown in DYGS liquid media (48 h incubation at 28°C under 180 rpm on an orbital shaker) with an optical density (600 nm) of 0.4 as a preinoculum for each assay, and the cell-free extracts were obtained by centrifugation (8,000 g, 5 min at 4°C). All determinations were performed in triplicate for each strain.

The IAA production was determined according to the Salkowski colorimetric assay as described by Sarwar and Kremer [37]; cell-free supernatants obtained from cultures in DYGS liquid media supplemented with tryptophan (100 μg ml$^{-1}$) and incubated for 48 h at 28°C and 180 rpm on an orbital shaker were used. The concentration of IAA produced by each strain was estimated after determining their absorbance at 530 nm with the aid of an IAA (Sigma Aldrich) standard curve. The protein content of the bacterial biomass resulting from the cell-free supernatants used in the Salkowski assay was determined by Bradford's method [38], and the IAA produced by each strain was expressed as the ratio of IAA (μg) in the supernatant per milligram of protein from the bacterial biomass.

The ability to solubilize inorganic phosphate was evaluated in liquid NBRIP media (10 ml) [39] supplemented with either iron phosphate ($FePO_4$) or aluminum phosphate ($AlPO_4$) each at 1.0 g L$^{-1}$ after incubation for seven days at 28°C and the determination of soluble phosphorus concentration by the phosphomolybdate method [40] in cell-free extracts. Flasks of noninoculated NBRIP were prepared under the same conditions to determine the amount of P that was spontaneously soluble under the experimental conditions. The P solubilization ability for each isolate was defined as the difference between the concentration of soluble phosphorus in

the respective NBRIP cell-free extract and the concentration of soluble phosphorus in the non-inoculated NBRIP.

The production of siderophores was determined in T-CAS media supplemented with 10% (v/v) CAS solution, where the development of a yellow or orange halo around bacterial colonies indicated the presence of siderophores [41,42]. Petri dishes with T-CAS media were seeded with two µL of a preinoculum suspension of each bacterial strain and incubated at 28˚C in the dark for five days. After the incubation period, the ratio between the diameter of the colored halo and the diameter of the bacterial colony, named the siderophore index (SI), was used as a quantitative estimate of siderophore production potential.

## Greenhouse trial

Inoculation trials using the same tomato and lulo genotypes previously used to isolate diazotrophic/N-scavenging bacteria were carried out under greenhouse conditions, aiming to quantify the ability of the isolates to promote the accumulation of biomass of plants subjected to N stress. Tomato and lulo seeds were surface disinfected [26] and sown into pots filled with 2 kg of unsterile sand, with five seeds per pot. At four (tomato) or seven (lulo) days after sowing, the seedlings were thinned to one healthy plant per pot, and the inoculation procedure was carried out by applying 1 ml of bacterial suspension to the seedlings. The bacterial suspensions were individually prepared for each isolated strain in DYGS liquid media (optical density of 0.4 at 600 nm) as described above. After the inoculation procedure, the pots were arranged as part of a randomized experimental design with five replicates, and the plants grew for another 40 (tomato) or 60 (lulo) days. Uninoculated plants were used as controls, and during the experimental period, the plants were watered with 100 ml of tap water twice per week and 100 ml of Hoagland's nutrient solution (with 10% N concentration) once per week. At harvest, the plants were dried to a constant weight at 65˚C following the determination of root dry weight and shoot dry weight (RDW and SDW, respectively).

## Statistical analysis

The plant growth-promoting traits and plant biometric parameters were subjected to one-way analysis of variance (ANOVA), and the means were compared by the Scott-Knott test ($p < 0.05$). In addition, Pearson's correlation analysis ($p < 0.05$) was carried out for the plant biometric data to explore the relationships between the studied parameters. All analyses were performed with the aid of R software (http://www.r-project.org) using the packages agricolae, ScottKnott and corrgram.

The strains were arbitrarily ranked for their potential to promote plant growth based on the bonitur scale [43]; the rankings encompassed all the data obtained from both the *in vitro* and *in vivo* characterizations. Briefly, the absolute value for each evaluated trait was converted to a percentage between the numeric value observed for a given strain and the mean trait value, considering only the positive results for the trait. The percentage of each PGP trait (IAA production, $FePO_4$ solubilization, $AlPO_4$ solubilization and siderophore production) was transformed into an arbitrary value ranging from zero to three according to the following scale: 0, trait not detected; 1, values lower than 35% of the mean trait value; 2, values between 35% and 70% of the mean trait value; and 3, values higher than 70% of mean trait value. The arbitrary values that were applied to the biometric parameters (SDW and PDW) ranged from zero to two according to the following scale: 0, values significantly lower than those observed for uninoculated plants ($p < 0.05$); 1, values similar to those of uninoculated plants; and 2, values higher than those observed for uninoculated plants. The maximum bonitur score for a given strain in the present study was 20.

## Results

### Bacterial isolates

The descriptive results of the number of isolates and population densities of the diazotrophic/ N-scavenging bacteria in the soils and unwashed roots of tomato and lulo are presented in Table 1. A total of 101 bacterial strains were found at population densities greater than 1 x $10^4$ cells g$^{-1}$ fresh weight of soil or unwashed *Solanum* roots. From this total, 26 isolates were obtained from soil samples, and 42 and 33 isolates were obtained from unwashed roots of tomato and lulo plants, respectively. Most isolates were collected using JMV culture media (49 isolates), while the LGI-P media yielded the fewest isolates (2 isolates). The number of diazo-troph/N scavenger bacterial strains varied with soil management conditions, with 40 strains associated with organic farming (both soil and unwashed roots), 34 strains isolated from sec-ondary forest samples and 27 strains obtained from samples under conventional farming.

The population densities of the diazotrophic/N-scavenging bacteria showed variations within the samples, within the culture media and under the soil management conditions, although no significant differences ($p < 0.05$) were found. The soil samples had, in general, lower population densities of diazotrophs/N scavenger bacteria than did the *Solanum* unwashed roots, with the exception of the MPNs from the ORG soil compared to those of the unwashed tomato roots from the same soil. Although the differences in population densities of diazotrophs/N scavenger bacteria were not significant, the *Solanum* plants grown in SF soils showed the greatest differences between bacterial populations from unwashed roots and the soils without plants. Considering the different semisolid N-free culture media, JMV (mannitol as a C source, pH 5.2) yielded the highest bacterial counts (mean MPN of 9.98 x $10^4$), followed by JNFb (malic acid, pH 5.8) and LGI (sucrose, pH 6.0); these last two had mean MPNs of 4.92 x $10^4$ and 2.85 x $10^4$, respectively. Samples associated with SF and ORG soils showed the high-est mean population densities of diazotrophs/N scavenger bacteria (21.22 x $10^4$ and 19.56 x $10^4$, respectively), while soils under conventional management had a lower mean MPN (12.22 x $10^4$) than did the other soils.

### Identification and comparative diversity analysis of isolated strains

The phylogenetic positioning of isolates based on partial 16S rRNA gene sequences revealed the dominance of the Proteobacteria phylum; specifically, three classes, Alphaproteobacteria, Gammaproteobacteria and Betaproteobacteria, constituted 61.4%, 19.8% and 18.8% of the total isolates, respectively (Fig 1). At the genus level, *Rhizobium* was predominant (59 isolate strains), followed by *Pseudomonas* (12 strains) and *Burkholderia* (11 strains), representing the Alpha-, Gamma- and Betaproteobacteria, respectively. Isolates of other genera present at low frequencies included *Caulobacter* and *Novosphingobium* (Alphaproteobacteria, one strain each), *Enterobacter* and *Stenotrophomonas* (Gammaproteobacteria, five and two strains, respectively), and *Cupriavidus* and *Variovorax* (Betaproteobacteria, three and four strains, respectively) (Fig 1A). The strains 04S, 12S and 16T were below the confidence threshold (95% identity) of the RDP classifier, and their identification at the genus level must be taken with caution because further analysis using BLAST against reference RNA sequences in the Gen-Bank database or classification in the SILVA ribosomal RNA database rendered different results. According to the RDP classifier, the isolates 04S, 12S and 16T belong to the genera *Xanthomonas* (70% confidence), *Rhizobium* (79% confidence) and *Pelomonas* (43% confi-dence), respectively, while the analysis through BLAST or SILVA positioned these isolates as *Stenotrophomonas* (99.53% identity according to BLAST), *Agrobacterium* (98.42% identity) and *Mitsuaria* (99.53% identity), respectively (Fig 1A). Although the results of the RDP

**Table 1. Number of isolates and population densities of diazotrophic/N-scavenging bacteria isolated via different semisolid N-free culture media from soils under different management conditions and from unwashed roots of tomato and lulo plants grown on those soils.**

| Isolation source[a] | | Semisolid N-free medium | | | | | | | | | | Total MPN[b] | |
|---|---|---|---|---|---|---|---|---|---|---|---|---|---|
| | | JMV | | NFb | | JNFb | | LGI | | LGI-P | | | |
| | | n[c] | MPN (x10⁴) | n | MPN (x10⁴) | n | MPN(x10⁴) (x104) (x10⁴) (x10⁴) | n | MPN (x10⁴) (x104) | n | MPN (x10⁴) | n | MPN (x10⁴) |
| Soil | SF | 5 | 7.67 ± 10.8 | 0 | 0 | 3 | 3.33 ± 3.5 | 0 | 0 | 0 | 0 | 8 | 11.00 |
| | CH | 0 | 0 | 0 | 0 | 5 | 7.67 ± 6.4 | 0 | 0 | 0 | 0 | 5 | 7.67 |
| | ORG | 10 | 16.00 ± 16.6 | 0 | 0 | 2 | 1.33 ± 2.3 | 1 | 1.33 ± 2.3 | 0 | 0 | 13 | 18.67 |
| Tomato | SF | 7 | 18.00 ± 23.8 | 2 | 3.00 ± 5.2 | 2 | 3.33 ± 0.6 | 2 | 2.00 ± 1.7 | 1 | 1.00 ± 1.7 | 14 | 27.33 |
| | CH | 9 | 9.00 ± 2.0 | 2 | 2.67 ± 2.3 | 4 | 4.00 ± 0.1 | 3 | 4.00 ± 0.1 | 0 | 0 | 18 | 19.67 |
| | ORG | 5 | 8.00 ± 3.6 | 0 | 0 | 5 | 6.33 ± 3.1 | 0 | 1.00 ± 1.7 | 0 | 0 | 10 | 15.33 |
| Lulo | SF | 5 | 10.0 ± 4.6 | 1 | 1.00 ± 1.7 | 4 | 8.33 ± 1.2 | 1 | 3.33 ± 0.6 | 1 | 2.67 ± 2.3 | 12 | 25.33 |
| | CH | 2 | 3.67 ± 0.6 | 0 | 1.00 ± 1.7 | 1 | 2.33 ± 2.1 | 1 | 2.33 ± 4.0 | 0 | 0 | 4 | 9.33 |
| | ORG | 6 | 5.67 ± 2.9 | 1 | 1.00 ± 1.7 | 5 | 7.67 ± 6.4 | 5 | 9.00 ± 6.0 | 0 | 1.33 ± 2.3 | 17 | 24.67 |
| | Total n[d] | 49 | | 6 | | 31 | | 13 | | 2 | | 101 | |

[a]SF, secondary forest soil with no agricultural use; CH, horticulture soil under conventional management; ORG, horticulture soil under organic management.

[b]The total population count is considered the sum of the MPNs from the semisolid N-free culture media for each sample source.

[c]Number of isolates retrieved.

[d]Total number of isolates obtained for each semisolid N-free culture medium considering all sample sources.

classifier were controversial for strains 04S, 12S and 16T, the identification of these and all other strains were based on the RDP classifier for further analysis and discussion throughout this study.

In addition to the quantitative analysis (Table 1), the qualitative data of diazotrophic/N-scavenging bacteria across the different isolation sources suggest higher diversity of this bacterial group in the soils compared to the unwashed *Solanum* roots (tomato or lulo, Fig 1B). Among the 11 total representative genera identified among the isolated strains, eight were

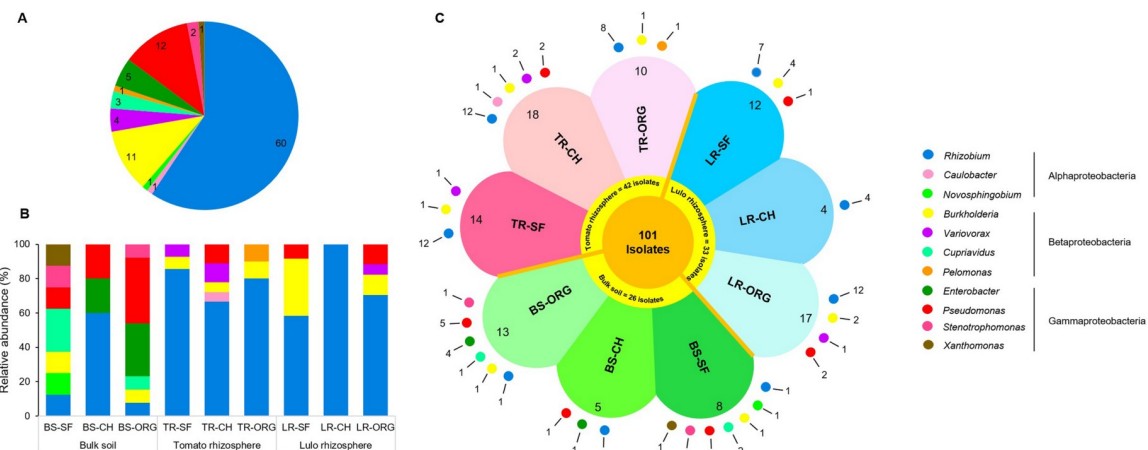

**Fig 1.** Qualitative and quantitative distribution of bacterial strains isolated from different sources and from soils under different management conditions according to phylogenetic positioning at the genus level (1A), relative abundance (1B) and isolation source (1C). The chart legend indicates the identified genus and respective classes. BS, soil; TR, tomato unwashed roots; LR, lulo unwashed roots; SF, secondary forest soil with no agricultural use; CH, horticulture soil under conventional management; ORG, horticulture soil under organic management.

found in soil samples from which five genera were not found, at least within the threshold population density, in association with *Solanum* roots (*Novosphingobium*, *Stenotrophomonas*, *Xanthomonas*, *Cupriavidus* and *Enterobacter*). The bacterial strains obtained from unwashed tomato roots represented six genera, where strains of one genus (*Pelomonas*) were obtained exclusively from these samples. In the case of strains isolated from lulo unwashed roots, four genera were represented, and in addition to those identified among strains isolated from tomato roots, two genera (*Variovorax* and *Caulobacter*) were exclusive to *Solanum* roots. The diversity of representative genera in relation to the environments studied (SF, CH and ORG) indicates the ubiquitous presence of *Pseudomonas*, *Burkholderia* and *Rhizobium*, although only *Rhizobium* was found among the strains from each sample (Fig 1B and 1C). It is noteworthy that the isolation frequency of *Rhizobium* strains was much higher from the unwashed roots of *Solanum* than from the soils, and representative strains of this genus were isolated using all five semisolid N-free culture media, albeit at higher frequencies using JNFb and JMV (38% and 37% of the isolates, respectively) (S2 Table).

The phylogenetic relationships among the isolated strains and the type species of their respective genera based on their 16S rRNA gene sequences are presented in the S1 File. These analyses indicated a high intrageneric diversity among the isolated strains, as indicated by the distribution of the 59 *Rhizobium* strains within 13 phylogenetic clusters, the 12 *Pseudomonas* strains distributed within 8 different clusters and the 11 *Burkholderia* strains distributed within five clusters. Among the 13 *Rhizobium* phylogroups, 10 of them housed two or more isolated strains, where the group with the higher number of strains (14 strains) clustered with *R. pusense* and presented strains obtained from different sources of isolation and from different soil management conditions; these isolates were obtained predominantly from tomato grown in ORG soil (strains 1T, 2T, 8T, 10T and 27T). The phylogenetic analysis of *Pseudomonas* strains resulted in three out of eight phylogroups presenting more than one strain and the clustering of isolates related to ORG (strains 20S, 11L and 25S) or CH soils (strains 14S and 17T), in addition to strains isolated from unrelated sources (strains 16S and 8L). The *Burkholderia* phylogroup with a relatively high number of isolates (four strains) was related to *B. metallica* and included strains related to ORG samples (strains 24S, 15L and 19L) and one strain related to SF samples (strain 1L).

## BOX-PCR genomic fingerprinting

The high diversity of bacterial strains studied was confirmed by the low number of clonal isolates demonstrated by BOX-PCR analysis (Fig 2). This high genotypic diversity is reflected in the 93 different fingerprint profiles at 95% similarity, which were arbitrarily clustered into 49 BOX groups with a 70% similarity cutoff to better present the results. Almost half of the BOX groups were formed by a single strain, while 25 BOX groups harbored two or more strains. In general, at 70% similarity, the BOX groups with multiple strains showed an uneven distribution of genotypes in relation to the isolation source (TR, LR or BS) and soil management type (SF, CH, ORG), although groups composed of strains somewhat related were also observed. The BOX groups g1, g5 and g20 comprised strains isolated from samples related to ORG soils; g1 comprised four *Rhizobium* strains isolated from tomato roots and a *Stenotrophomonas* strain isolated from the soil; g5 presented one *Rhizobium* strain isolated from tomato roots and two *Rhizobium* strains from lulo roots; and g20 comprised two *Burkholderia* strains isolated from lulo roots.

In this sense, the BOX groups g19 and g25 comprised strains isolated from samples related to the SF soil, and g24 comprised strains isolated from samples related to CH. The g19 group comprises six *Rhizobium* strains isolated from tomato roots, g25 comprises two *Rhizobium*

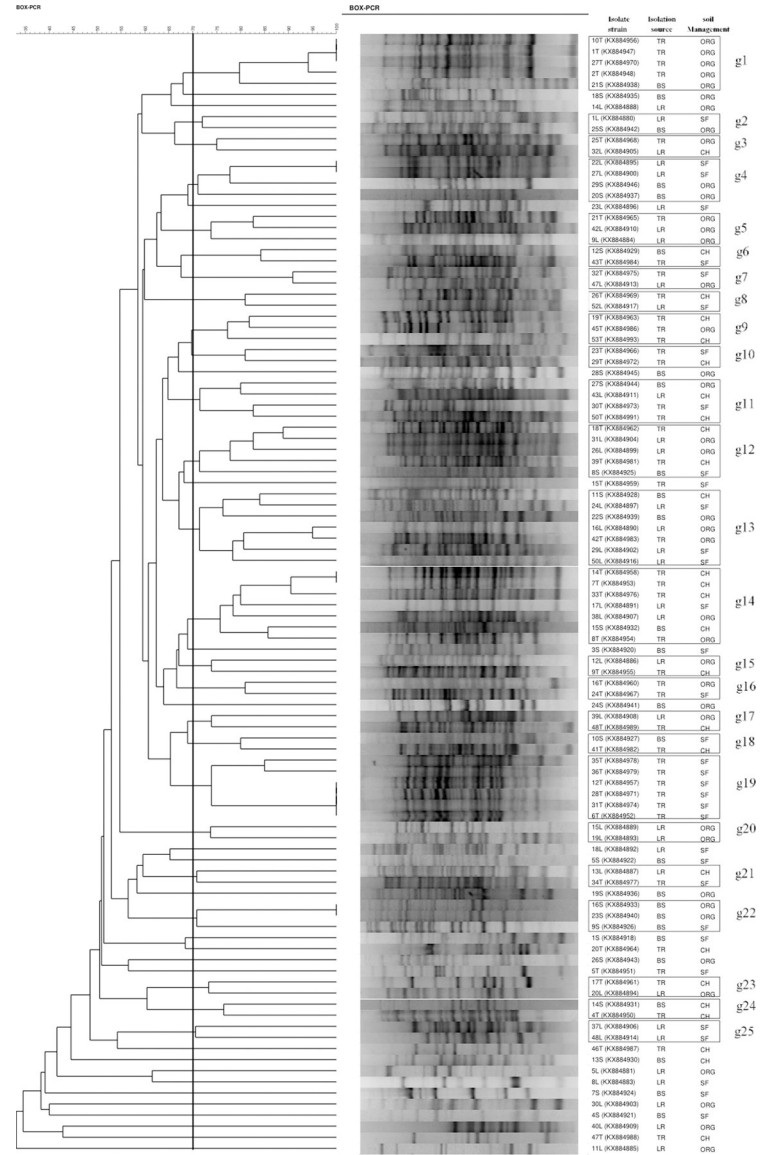

**Fig 2. Dendrogram representing the genotype diversity and genetic relationships estimated by the cluster analysis of BOX-PCR fingerprints of diazotrophic/N-scavenging bacteria isolated from different sources and from soils under different management conditions.** Dendogram was constructed using the Jaccard coefficient (2% tolerance in terms of band size) with the UPGMA algorithm. Isolation sources: BS (soil); LR (lulo unwashed roots), TR (tomato unwashed roots). Soil management conditions: CH (horticulture under conventional management); ORG (horticulture under organic management); SF (secondary forest with no agricultural use).

strains isolated from lulo roots, and g24 comprises a *Pseudomonas* strain isolated from the soil and a *Rhizobium* strain isolated from tomato roots. Considering the distribution of BOX profiles in the groups according to isolation source, it was clear that groups with strains isolated from TR (g9, g10, g16 and g19), LR (g20 and g25) or BS samples (g22) included strains isolated from soils under different management types (g9, g10, g16 and g22). It is noteworthy that the definition of 70% similarity adopted for group the BOX fingerprints was enough to partially discriminate strains at the genus level. Twelve out of 25 groups harbored phylogenetically

close strains: 10 groups with only *Rhizobium* strains and one group each with *Burkholderia* or *Pseudomonas* strains. The clonal strains identified by BOX analysis were identified mostly as *Rhizobium* species (strains 10T, 1T and 27T in g1; strains 22L and 27L in g4; strains 14T and 7T in g14; and strains 12T, 28T, 31T and 6T in g19) or *Pseudomonas* species (strains 16S and 23S atg22) and were all confluent with the isolation source and soil management type.

## Plant growth-promoting traits

Fig 3 summarizes the qualitative results of the biochemical characterization of the isolated strains, which are quantitatively presented in S3 Table and S1 Fig. Of the 101 PGPBs isolated, 91 showed at least one growth-promoting trait, and 68 showed two or more traits (Fig 3A). The ability to synthesize IAA was observed in 49 strains, accounting for 22 isolates from tomato roots, 16 isolates from lulo roots and 11 isolates from the soil (Fig 3B). The IAA-producing isolates were identified as *Rhizobium* (37 strains), *Pseudomonas* (six strains) and *Enterobacter* (five strains), as well as *Burkholderia* (one strain) (Fig 3C). The majority of strains (72) presented $FePO_4$-solubilizing activity, of which 36 of these strains were isolated from TR samples; further, 44 strains showed $AlPO_4$-solubilizing activity (Fig 3C), with an almost even distribution between the different isolation sources (BS, TR and LR). Furthermore, the quantitative results for P solubilization indicated that $FePO_4$ solubilization ranged from 0.4 mg g$^{-1}$ to 56 mg g$^{-1}$ (*Enterobacter* sp. strain 15S), with a mean value (considering the data from all positive strains) of 6.36 mg g$^{-1}$, while the solubilization of $AlPO_4$ ranged from 0.62 mg g$^{-1}$ to 17.05 mg g$^{-1}$, with a mean value of 3.85 mg g$^{-1}$ (*Burkholderia* sp. strain 23L). When the T-CAS medium was used to grow the strain collection, 44 strains showed the ability to produce siderophores, with siderophore index (SI) values varying from 1.06 to 3.23 (*Burkholderia* sp. strain 1L); the best siderophore-producing strains were *Rhizobium*, *Pseudomonas* and *Burkholderia* representatives. A joint evaluation of all four PGP traits revealed seven strains with competence to produce positive results for IAA, P solubilization of both insoluble sources and siderophore production, comprising five *Rhizobium* (strains 21T, 29T, 41T, 37L and 39L), one *Pseudomonas* (strain 17T) and one *Burkholderia* (strain 15L). Further, ten strains out of the 101 studied showed negative results for all of these same PGP traits evaluated *in vitro*: *Rhizobium* sp. strains 12T, 9L, 13L and 38L; *Cupriavidus* sp. strain 10S; *Novosphingobium* sp. strain 5S; *Pseudomonas* sp. strain 8L, *Stenotrophomonas* sp. strain 1S; *Variovorax* sp. strain 30L; and *Xanthomonas*/*Stenotrophomonas* sp. strain 4S.

## Growth-promoting effects of isolates on tomato and lulo under greenhouse conditions

To validate the potential of the diazotrophs/N scavenger strains to promote the growth of tomato and lulo plants under N-limiting conditions, a greenhouse trial was conducted, and the effects of inoculation on plant biomass accumulation are presented in Fig 4 and S3 Table. Inoculation of *Rhizobium* (strains 35T and 36T), *Pseudomonas* (strains 16S and 9S), *Enterobacter* (strain 19S), *Cupriavidus* (strain 26S), *Burkholderia* (strain 5T) and *Variovorax* (strain 7S) increased the RDW of tomato plants compared to uninoculated plants (Fig 4A), and significant decreases in RDW were indeed detected as a result of inoculation by most of the strains (60 strains). The strains with positive effects on tomato root biomass were isolated mainly from soils without plants (5 strains) or soils under SF management conditions (5 strains). Interestingly, none of the bacterial strains that promoted increases in tomato RDW resulted in significant increases in the SDW of tomato plants; in fact, in addition to 36 other strains, four strains that increased the tomato RDW (strains 9S, 5T, 35T and 36T) also significantly decreased the tomato SDW (Fig 4B). Positive inoculation effects on tomato SDW were

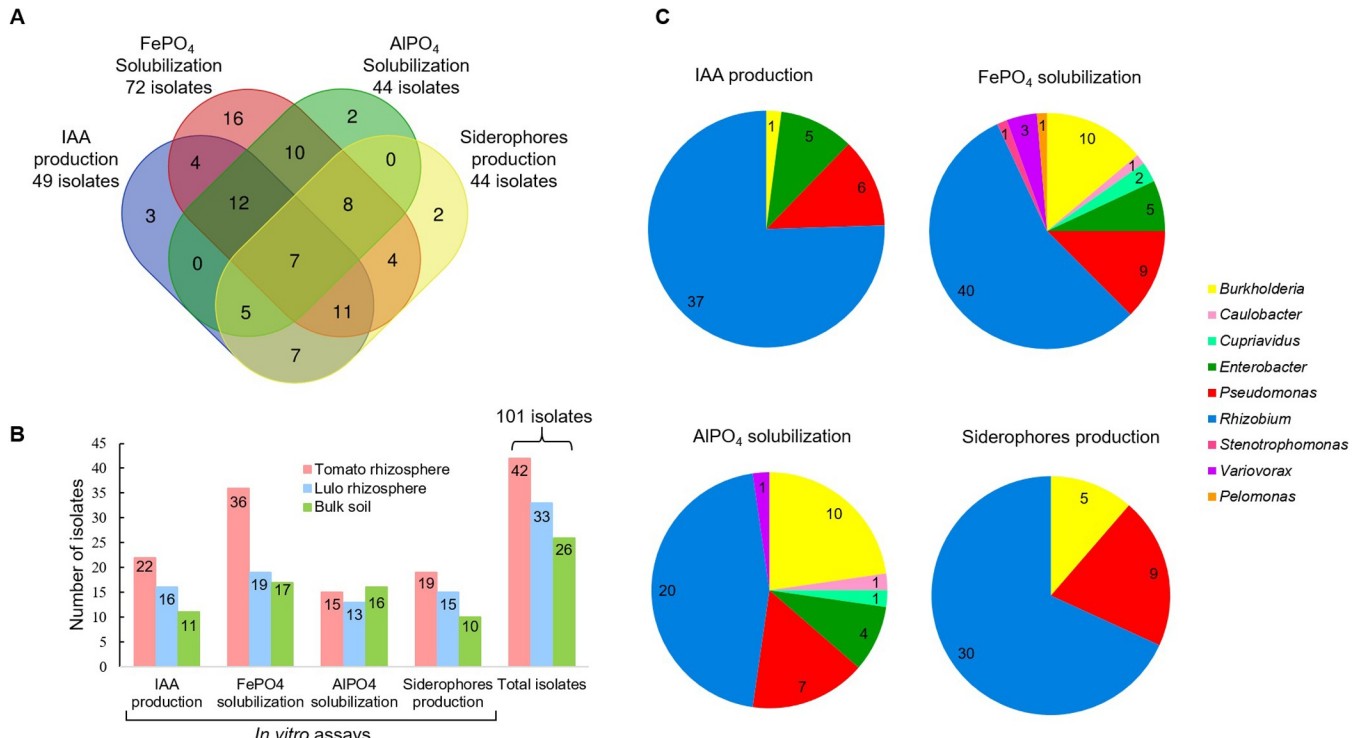

**Fig 3. Qualitative representation of plant growth-promoting traits observed for diazotrophic/N-scavenging bacteria isolated from different sources and from soils under different management conditions.** (**A**) Venn diagram showing unique and shared plant growth-promoting (PGP) traits detected for the bacterial strains studied. (**B**) Quantitative distribution of bacterial strains according to isolation source and PGP traits. (**C**) Quantitative distribution of bacterial strains according to phylogenetic position at the genus level and PGP traits.

observed for 20 strains, but despite this, 14 of these strains induced decreases in RDW. The strains isolated from tomato roots and identified as *Rhizobium* spp. were predominant among the SDW-promoting isolates for tomato (12 strains), and regardless of the phylogenetic identification, 14 strains were autochthonous (isolated from tomato roots).

The strains that showed the best increases in tomato RDW were *Rhizobium* sp. 36T, *Cupriavidus* sp. 26S and *Pseudomonas* sp. 16S, while the strains that induced relatively high increases in tomato SDW were *Stenotrophomonas* sp. 1S and *Rhizobium* sp. strains 8T and 21T. It is noteworthy that all bacterial strains that increased the root biomass of tomato plants showed the ability to solubilize $FePO_4$, and five were also able to solubilize $AlPO_4$, while both IAA and siderophore production was observed for two out of the eight strains. The ability to solubilize $FePO_4$ also prevailed among strains that promoted significant increases in the SDW of tomato (75% of growth-promoting strains); $AlPO_4$ solubilization was observed for 75% and 30% of strains that increased the RDW and SDW of tomato, respectively. IAA biosynthesis and siderophore production were observed in 37.5% and 25% of the strains that increased the tomato RDW, respectively, while considering the strains that increased SDW, both traits occurred in 55% of the strains. Interestingly, three strains for which none of the four studied PGP traits were identified caused significant increases in the shoot biomass of tomato (*Stenotrophomonas* sp. strain 1S, *Rhizobium* sp. strain 12T and *Xanthomonas* sp. strain 04S).

Inoculation of diazotrophic/N-scavenging bacteria in lulo (*S. quitoense*) was apparently more effective than that in tomato (*S. lycopersicum*) according to the high number of strains that resulted in increases in RDW or SDW and the low number of strains with negative effects on these parameters. A positive inoculation response on the RDW of lulo was observed for 50

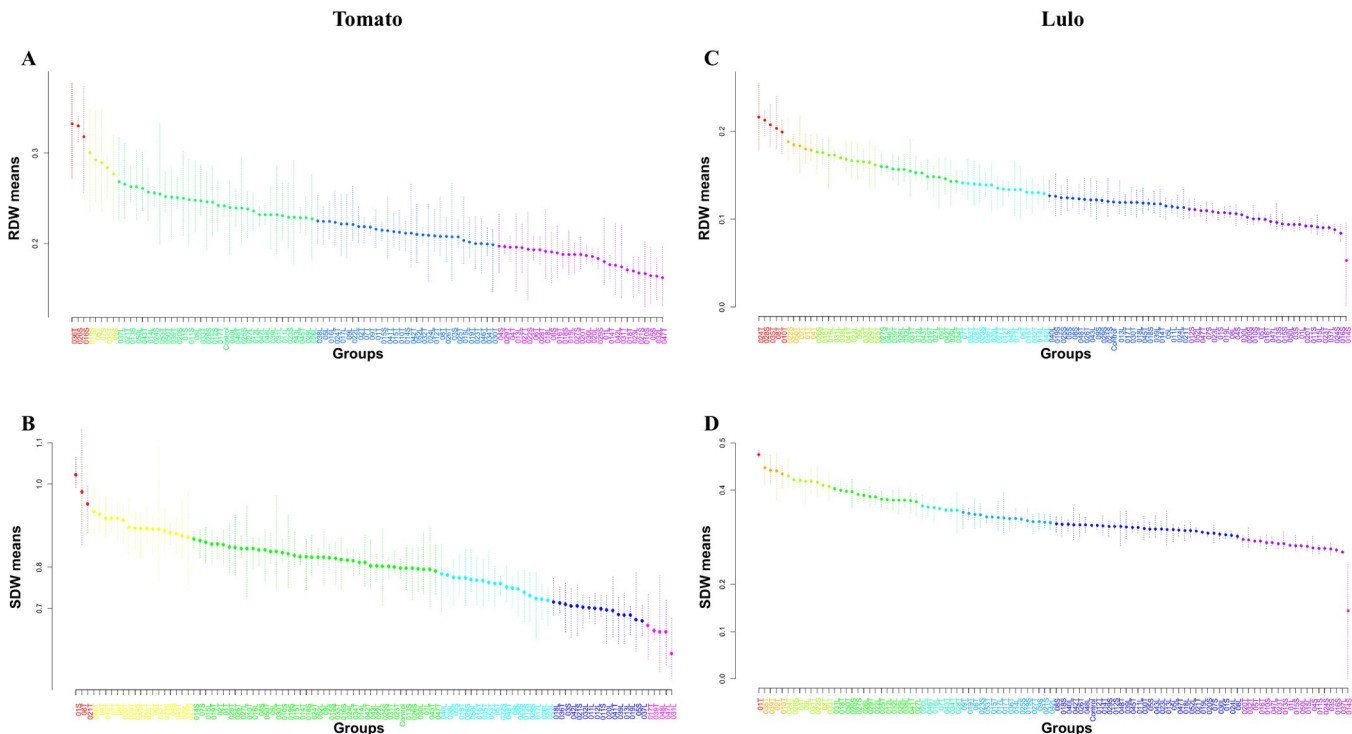

**Fig 4. Accumulation of biomass of the root system or aerial tissues of lulo and tomato plants grown under N-limiting conditions in response to the inoculation of 101 different diazotrophic/N-scavenging bacteria. (A, C) Root dry weight (g plant$^{-1}$) of tomato (*S. lycopersicum*) and lulo (*S. quitoense*) plants, respectively. (B, D) Shoot dry weight (g plant$^{-1}$) of tomato (*S. lycopersicum*) and lulo (*S. quitoense*) plants, respectively.** The mean values plotted in the same color represent groups that do not significantly differ at $p < 0.05$ according to the Scott-Knott algorithm. The bars refer to the maximum and minimum values for each plot. RDW, root dry weight; SDW, shoot dry weight.

different strains (Fig 4C); of which 42 were identified as *Rhizobium*, three were identified as *Enterobacter*, and two were identified as *Variovorax*, plus one each of *Burkholderia*, *Cupriavidus*, and *Pseudomonas*. Most strains that increased the lulo RDW were isolated from unwashed tomato roots (30 strains) or were associated with ORG management conditions (21 strains). Furthermore, 51 strains caused increases in the SDW of lulo plants (Fig 4D); these strains comprised 38 *Rhizobium* strains, four strains each of *Pseudomonas* and *Enterobacter*, two *Variovorax* strains, one *Burkholderia* strain, one *Cupriavidus* strain and one *Stenotrophomonas* strain. Most strains with positive effects on lulo SDW were isolated from tomato roots (28 strains) or were related to ORG management conditions (23 strains).

As observed for the tomato growth-promoting strains, the PGP trait associated with the highest frequency among the lulo growth-promoting bacteria was FePO$_4$ solubilization, which was identified in 76% and 78% of the strains that increased the RDW and SDW of lulo, respectively. Following the qualitative analysis of traits observed for the strains that promoted the RDW and SDW of lulo, IAA biosynthesis was found in approximately 52%, followed by the production of siderophores and AlPO$_4$ solubilization (approximately 40% and 35% of lulo growth-promoting strains, respectively). In contrast to the performance of the inoculated strains in tomato, negative inoculation effects on lulo SDW by the same strain that increased the RDW were observed only for *Rhizobium* sp. 32T, and none of the strains that increased the lulo SDW induced decreases in RDW. However, 46 strains induced increases in both the RDW and SDW of lulo plants. Greater increases in the RDW of lulo were observed in plants inoculated with the *Rhizobium* sp. strains 10T, 24T, 35T and 8T, as well as *Enterobacter* sp.

strain 28S; considering the SDW, inoculation of *Rhizobium* sp. strain 1T resulted in the greatest biomass accumulation. In addition, two *Rhizobium* sp. strains that were negative for the four PGP traits evaluated caused increases in the RDW of lulo: 12T and 38L, which also increased the SDW of lulo.

Overall, the inoculation results indicate that strains 26S, 35T and 36T induced increases in the RDW of both *Solanum* species, while strains 4T, 6T, 7T, 8T, 9T, 25T, 27T, 29T, 46T, 29L, 23S and 27S increased the SDW of these species. Excluding the strains that showed a significant decrease in tomato RDW or SDW, as noted above, three strains had a positive effect on the biomass accumulation in both tomato and lulo under N-limiting conditions. The first was *Pseudomonas* sp. strain 23S, which was isolated from ORG soil and showed potential to solubilize $AlPO_4$ and $FePO_4$ and produce siderophores, resulting in increases in the SDW of tomato and lulo as well as an increase in the RDW of lulo plants. The second was *Cupriavidus* sp. strain 26S, which was isolated from ORG soil, showed potential for solubilizing $AlPO_4$ and $FePO_4$ and induced increases in the RDW of tomato and lulo as well as increases in the SDW of lulo. The third was *Rhizobium* sp. strain 29L, which was isolated from the unwashed roots of lulo plants grown in CH soil and exhibited potential to synthesize IAA and solubilize $AlPO_4$ and $FePO_4$ phosphates, in addition to promoting increases in the RDW of tomato and in both the RDW and SDW of lulo. A quantitative distribution of bacterial strains with unique and shared positive effects on both the shoot and root biomass of inoculated tomato and lulo plants is shown in S2 Fig, which reinforces that none of the bacterial strains showed the capability to increase the root and shoot biomass of both tomato and lulo. Furthermore, the inoculation of diazotrophic/N-scavenging bacteria in tomato suggests a competitor effect on increases in root and shoot biomass, because the isolates that increased the RDW had negative or no effects on the SDW and vice versa. On the other hand, inoculation of diazotrophic/N-scavenging bacteria in lulo showed a positive correlation between RDW and SDW, with 31 bacterial strains inducing significant increases in both RDW and SDW (S2 Fig).

## Assessment of the biotechnological potential of the isolates showing *in vitro* and *in vivo* PGP traits

The collection of bacterial strains was ranked for their apparent plant growth-promoting potential to facilitate the selection of candidates for future inoculation trials under field conditions. Based on the bonitur scale [43], a bacterial strain was considered to have potential for additional biotechnological studies if its rank was greater than or equal to 10 out of 20 maximum points; 24 strains indicating biotechnological potential were revealed according to our assessment (Table 2). A complete list of ranked strains is shown in S3 Table. These promising isolates are represented by the genera *Rhizobium* (12 strains), *Enterobacter* (5 strains), *Pseudomonas* (5 strains), *Burkholderia* (1 strain) and *Cupriavidus* (1 strain), from which 11 strains were isolated each from BS or TR and from which the remaining 2 strains were isolated from LR. The top five bacterial strains with biotechnological potential were *Enterobacter* sp. strains 19S, 27S and 28S, all of which were isolated from BS-ORG samples, and *Rhizobium* sp. strains 21T and 4T, which were isolated from TR-ORG and TR-CH, respectively. A correlation analysis of the biochemical characterization of the strains and the biometric parameters of the inoculated plants indicated significant and positive correlations between them and the bonitur scale, although the correlation values were below 50% (S3 Fig). In addition, significant negative correlations between biochemical ($AlPO_4$ solubilization and siderophore production) and biometric parameters (RDW and SDW) were evidenced for lulo.

**Table 2. Rank position and arbitrary values converted from the percentage (%) between the absolute value and the mean trait value (*in vitro* and *in vivo* determinations) for diazotroph/N scavenger bacterial strains considered to have high biotechnological potential to promote the growth of *Solanum* (rank greater than or equal to 10) according to the bonitur scale.**

| Isolate ID | Isolation source[a] | Genus[b] | Plant growth-promoting traits | | | | | | | | Bonitur score[k] | Rank |
|---|---|---|---|---|---|---|---|---|---|---|---|---|
| | | | *In vitro* assays | | | | *In vivo* assays: tomato | | *In vivo* assays: lulo | | | |
| | | | IAA[c] | FePO$_4$[d] | AlPO$_4$[e] | Sider[f] | RDW[g] | SDW[h] | RDW[i] | SDW[j] | | |
| 019S | BS-ORG | *Enterobacter* | 3 | 3 | 2 | 0 | 2 | 1 | 1 | 2 | 14 | 1st |
| 021T | TR-ORG | *Rhizobium* | 1 | 2 | 2 | 2 | 1 | 2 | 1 | 1 | 12 | 2nd |
| 027S | BS-ORG | *Enterobacter* | 3 | 3 | 0 | 0 | 0 | 2 | 2 | 2 | 12 | 2nd |
| 028S | BS-ORG | *Enterobacter* | 3 | 3 | 1 | 0 | 1 | 0 | 2 | 2 | 12 | 2nd |
| 04T | TR-CH | *Rhizobium* | 3 | 1 | 0 | 2 | 0 | 2 | 2 | 2 | 12 | 2nd |
| 014L | LR-ORG | *Rhizobium* | 2 | 0 | 1 | 2 | 1 | 1 | 2 | 2 | 11 | 3rd |
| 017T | TR-CH | *Pseudomonas* | 1 | 3 | 1 | 2 | 1 | 0 | 1 | 2 | 11 | 3rd |
| 022S | BS-ORG | *Enterobacter* | 3 | 3 | 1 | 0 | 0 | 0 | 2 | 2 | 11 | 3rd |
| 023S | BS-ORG | *Pseudomonas* | 0 | 1 | 2 | 2 | 0 | 2 | 2 | 2 | 11 | 3rd |
| 026S | BS-ORG | *Cupriavidus* | 0 | 1 | 3 | 0 | 2 | 1 | 2 | 2 | 11 | 3rd |
| 027L | LR-SF | *Rhizobium* | 2 | 1 | 0 | 2 | 1 | 1 | 2 | 2 | 11 | 3rd |
| 029T | TR-CH | *Rhizobium* | 1 | 1 | 1 | 2 | 0 | 2 | 2 | 2 | 11 | 3rd |
| 035T | TR-SF | *Rhizobium* | 3 | 1 | 1 | 0 | 2 | 0 | 2 | 2 | 11 | 3rd |
| 036T | TR-SF | *Rhizobium* | 0 | 3 | 0 | 2 | 2 | 0 | 2 | 2 | 11 | 3rd |
| 041T | TR-CH | *Rhizobium* | 3 | 1 | 1 | 2 | 0 | 0 | 2 | 2 | 11 | 3rd |
| 09S | BS-SF | *Pseudomonas* | 1 | 2 | 3 | 0 | 2 | 0 | 1 | 2 | 11 | 3rd |
| 015S | BS-CH | *Enterobacter* | 3 | 3 | 3 | 0 | 0 | 1 | 0 | 0 | 10 | 4th |
| 016S | BS-ORG | *Pseudomonas* | 0 | 2 | 3 | 2 | 2 | 1 | 0 | 0 | 10 | 4th |
| 024S | BS-ORG | *Burkholderia* | 0 | 2 | 3 | 2 | 1 | 2 | 0 | 0 | 10 | 4th |
| 025S | BS-ORG | *Pseudomonas* | 0 | 2 | 1 | 2 | 1 | 1 | 1 | 2 | 10 | 4th |
| 025T | TR-ORG | *Rhizobium* | 1 | 2 | 0 | 2 | 0 | 2 | 1 | 2 | 10 | 4th |
| 02T | TR-ORG | *Rhizobium* | 1 | 2 | 0 | 2 | 0 | 1 | 2 | 2 | 10 | 4th |
| 050T | TR-CH | *Rhizobium* | 1 | 2 | 1 | 0 | 1 | 1 | 2 | 2 | 10 | 4th |
| 08T | TR-ORG | *Rhizobium* | 2 | 0 | 0 | 2 | 0 | 2 | 2 | 2 | 10 | 4th |

[a]BS, soil; LR, lulo unwashed roots; TR, tomato unwashed roots; CH, horticulture soil under conventional management; ORG, horticulture soil under organic management; SF, secondary forest soil with no agricultural use.

[b]According to the RDP classifier.

[c]IAA, indole-3-acetic acid scores: 0, IAA below the detection limit; 1, $\leq 21.5$ µg IAA mg$^{-1}$ protein; 2, $\geq 21.5$ and $\leq 44.7$ µg IAA mg$^{-1}$ protein; 3, $\geq 44.7$ µg IAA mg$^{-1}$ protein.

[d]FePO$_4$ solubilization scores: 0, FePO$_4$ solubilization below the detection limit; 1, $\leq 4.1$ mg g$^{-1}$; 2, $\geq 4.1$ mg g$^{-1}$ and mg g$^{-1}$ $\leq 8.4$; 3, $\geq 8.4$ mg g$^{-1}$.

[e]AlPO$_4$ solubilization scores: 0, AlPO$_4$ solubilization below the detection limit; 1, $\leq 2.5$ mg g$^{-1}$; 2, $\geq 2.5$ mg g$^{-1}$ and $\leq 5.2$ mg g$^{-1}$; 3, $\geq 5.2$ mg g$^{-1}$.

[f]Siderophore index (SI) scores (ratio of colored halo Ø:colony Ø): 0, no visible colored halo in T-CAS media; 1, $\leq 1.1$ SI; 2, $\geq 1.1$ SI and $\leq 2.4$ SI; 3, $\geq 2.4$ SI.

[g]RDW, root dry weight scores for tomato: 0, $\leq 0.23$ g plant$^{-1}$; 1, $\geq 0.23$ g plant$^{-1}$ and $\leq 0.27$ g plant$^{-1}$; 2, $\geq 0.27$ g plant$^{-1}$.

[h]SDW, shoot dry weight scores for tomato: 0, $\leq 0.79$ g plant$^{-1}$; 1, $\geq 0.79$ g plant$^{-1}$ and $\leq 0.87$ g plant$^{-1}$; 2, $\geq 0.87$ g plant$^{-1}$.

[i]Root dry weight scores for lulo: 0, $\leq 0.11$ g plant$^{-1}$; 1, $\geq 0.11$ g plant$^{-1}$ and $\leq 0.13$ g plant$^{-1}$; 2, $\geq 0.13$ g plant$^{-1}$.

[j]Shoot dry weight scores for lulo: 0, $\leq 0.30$ g plant$^{-1}$; 1, $\geq 0.30$ g plant$^{-1}$ and $\leq 0.33$ g plant$^{-1}$; 2, $\geq 0.33$ g plant$^{-1}$.

[k]Sum of all assessment scores.

# Discussion

As the world faces a growing population subjected to limited natural resource availability as well as growing pressure from climate change over food and feed production, effective and environmentally friendly solutions driven to ensure food safety and energy supplies must be

identified and implemented. Current agricultural practices are responsible for large amounts of greenhouse gas emissions (GGE) resulting from the use of oil-based inputs in large and increasing amounts, and environmental harm can potentially occur from the misuse of agrochemicals [44]. Conversely, natural mechanisms involved in plant nutrition and plant protection against biotic and abiotic stresses can be associated with plant growth-promoting bacteria (PGPB), which have been shown to be a sustainable alternative as substitutes, at least in part, for the use of agrochemicals [45,46]. In this context, microbiological culture-based approaches for the search of plant-associated bacteria with PGP functions and functional and ecological studies of these microbial groups can provide a basis for the development of novel biological inputs for agriculture [6]. The present manuscript followed this route to partially reveal the biodiversity of diazotrophic/N-scavenging bacteria that are associated with different *Solanum* species (tomato and lulo) and soils under different management conditions and that are found at population densities greater than $1x10^4$ cells $g^{-1}$ (in both plant tissue and the soil).

The isolated bacterial strains were previously treated in this study as diazotrophs or N scavengers because N-free media allow the growth of bacteria with the ability to scavenge traces of nitrogen sources such as $NH_3$ and $N_2O$ from the atmosphere [47] and because nitrogenase activity and the expression of its related genes were not assessed in the isolated strains. Nevertheless, it is noteworthy that nitrogen fixation ability, or at least the presence of structural nitrogenase genes, has been reported for all the genera identified in the present study except for *Xanthomonas* sp. strain 4S, whose identification diverged between *Xanthomonas* and *Stenotrophomonas* according to the RDP classifier and BLAST/SILVA database information, respectively. Culture-based approaches using N-free culture media have been successfully used to estimate populations of diazotrophs in association with different plant species and environmental conditions [27,48–51]. Although they are in limited range for proper descriptions of microbial community structures in complex environments such as soils and plant roots, culture-dependent methods are of great importance for detecting low-abundance microbial phylotypes and for performing intense studies of the ecological and physiologic roles of specific strains in a given ecosystem [52–54].

As holobionts, individual plants have evolved to interact with microorganisms and somehow developed mechanisms to increase the population size of specific microbial groups in the vicinity of the roots; these microbes are recruited from the soil microbiome as part of a phenomenon known as the rhizosphere effect [55–57]. Furthermore, different soils and plant genotypes, in addition to the chosen method to assess the biodiversity of representative bacteria from soil and root ecosystems, may result in different pictures of the structure and composition of microbial communities associated with plants [52,58,59]. In the present work, the use of N-free semisolid culture media to assess diazotrophic/N-scavenging bacteria revealed variations in the isolation frequency of the targeted microbial groups according to the isolation source (BS, TR or LR) and the native microbial community (SF, CH or ORG soils). The rhizosphere effect imposed by different *Solanum* species suggests a preference of both plant species to increase the population size of representative *Rhizobium* strains in the rhizosphere, at least under the tested conditions. This was evidenced by a higher isolation frequency of *Rhizobium* in the TR and LR samples compared to the BS samples, regardless of the soil management conditions under which the plants were grown and the type of N-free semisolid culture medium used in the isolation procedure. Rhizobia are best known to develop symbiotic interactions with legumes, where mutualistic relationships progress to modifications of the root architecture of the host plant by the formation of nodules where nitrogen fixation occurs [60]. Nevertheless, nonsymbiotic and even parasitic rhizobial strains can be found in abundance in rhizosphere and nonrhizospheric soils, and the resulting plant-rhizobium interaction can be determined by the genetic content of both partners and environmental pressure [61–64].

In fact, the pool of *Rhizobium* species comprises several representatives originally isolated from samples other than legume nodules, including endophytes and rhizosphere-colonizing species isolated from plants in the Araceae, Asteraceae, Convallariaceae, Poaceae, Rosaceae, and Solanaceae families (www.bacterio.net/rhizobium.html). As stated by Berge et al. [65], the variability of environments and activities observed among species of rhizobia indicates the involvement of this group in "a broad range of functions in diverse ecosystems". The findings reported here indicate that *Rhizobium* species can be selectively increased in population size in association with tomato and lulo plants, as evidenced by the increased isolation frequency of *Rhizobium* strains from plants grown in soils with no history of legume cropping, such as the SF soil. Several studies have demonstrated that *Rhizobium* can develop associative relationships with nonleguminous plants with the potential for used as PGPB, although the molecular basis of such interactions is still poorly understood [66–71]. However, colonization of *Solanum lycopersicum* by *Rhizobium* has been scarcely reported, and the bacteriome of *S. quitoense* remains unexplored. Studies based on NGS have shown the presence of Rhizobiales as a component of the tomato microbiome, even as the main phylogenetic group in the endosphere of roots [72–75]. Furthermore, Pseudomonadales, Enterobacteriales, Rhizobiales, Burkholderiales and Xanthomonadales have been described as predominant Proteobacteria orders that colonize tomato roots [76], while several diazotrophic *Burkholderia* species were found in tomato plants grown in fields in Mexico [77].

The question raised in response to the high frequency of *Rhizobium* species isolated from *Solanum* plants observed in the present work is whether these strains are components of the native soil microbiome or are derived from anthropogenic environmental interventions, such as legume cropping or the use of commercial rhizobial inoculants. According to the phylogenetic relationships between the representative *Rhizobium* isolates and the *Rhizobium* type strains, we observed isolates that clustered close to rhizobial species that are commonly used as inoculants (*R. freirei* cluster), although such strains encompassed approximately 6% to 29% of the total rhizobia isolates depending on the stringency adopted to consider distinct phylogenetic clusters. Further, the high genomic diversity revealed by BOX-PCR fingerprinting suggests that the isolates do not represent a homogeneous evolved group, as would be expected if they were related to human-selected strains for use as inoculants. The phylogenetic positioning of the *Rhizobium* strains isolated from soils (strains 03S, 11S, 13S and 18S) away from the *R. freirei* cluster also suggests that native soils were not heavily colonized by strains used as commercial inoculants. The genetic diversity and community structure of rhizobia, which are natural microsymbionts of legumes, are commonly studied using a given leguminous plant as a trap plant, following the isolation of bacteria from nodules and their further characterization (genetic, functional, and phenotypic characterization). Such studies suggest that selective enrichment of rhizobial strains from the pool of the rhizobial community in the soil occurs based on symbiotic compatibility (gene background) with the plant, and depending on the plant used, the biodiversity of isolates can be restricted [78,79].

Attempts to identify efficient plant growth-promoting bacteria from representative isolates of a given plant microbiome generally begin with the *in vitro* characterization of PGP traits exhibited by the candidate strain, although this method is controversial because the bacterial metabolism is modulated according to environmental conditions, including plant feedback [80,81]. Nevertheless, significant correlations between *in vitro* bacterial traits and the respective inoculation response are commonly reported in the literature, including those used to suggest which mechanism of growth-promoting is predominant in plant-bacterium associations [82–84]. From the set of direct mechanisms that PGPB can use to enhance plant growth, the biosynthesis of phytohormones such as IAA has been reported to play a major role due to its effects on root architecture and, consequently, on the acquisition of water and nutrients by

plants [82,85–87]. The ability to solubilize P forms has recently been raised as a target trait for bacterial inoculants because, in most soils, the soluble P amount is low and because phosphates are a nonrenewable mineral resource needed for plant nutrition [88,89]. Most P-solubilizing bacteria act on inorganic secondary P minerals such as calcium, iron and aluminum phosphates by releasing adsorbed P by the production of organic acids or by the mineralization of organic P forms [90]. Another important trait from associative bacteria involved in the promotion of plant growth is the production of siderophores, which are believed to play a role in direct and indirect mechanisms of growth promotion by increasing iron availability for plant nutrition and by reducing its availability to pathogens [91–93].

In the present work, no positive or significant correlation resulted between the PGP traits determined *in vitro* and the plant biometric parameters; instead, significant and negative correlations resulted between the lulo growth and the $AlPO_4$ solubilization and siderophore production by diazotrophic/N-scavenging bacteria. As $AlPO_4$ is described as a highly insoluble chemical P form, bacterial metabolism related to $AlPO_4$ solubilization may also implicate a strong decrease in the pH of the colonization site. Although the pH of the solutions used to test for $AlPO_4$ solubilization was not addressed in this work, a *Pseudomonas* sp. strain described as effective at solubilizing $AlPO_4$ can reduce the pH of culture media from 7,0 to below 4,0 in two days, without producing organic acids [94]. Considering that most strains with a positive effect on biomass accumulation in tomato have shown potential to solubilize $FePO_4$, this trait is suggested to be relevant for the observed responses. Furthermore, while some strains for which none of the PGP traits were observed (P solubilization, IAA and siderophore production) caused increased tomato and lulo biomass, other PGP traits such as N supply may have also played important roles in the observed inoculation effects, even more so considering that plants were under N starvation. In fact, due to the complexity of the biotic and abiotic interactions that influence plant growth and physiology and, consequently, the plant-plant growth-promoting bacterium molecular crosstalk, in addition to the variability of phylogenetic groups and their respective metabolic needs, the identification of mechanisms involved in plant growth promotion by a given plant growth-promoting bacterial strain is not trivial [22].

A comparative analysis of plant growth-promoting bacterial genomes carried out by Cai et al. [95] highlighted common genetic features of 151 plant-associative bacteria (both pathogenic and beneficial strains), supporting the hypothesis for the convergent evolution of different bacterial taxa to select genes involved in the colonization of plant hosts and development of compatible (specific) interactions. Although limited in range over the bacterial biodiversity of plant microbiomes, the genomic diversity, isolation frequency and population sizes of diazotrophic/N-scavenging bacteria presented in our study demonstrate that soil communities differ from those in association with *Solanum*, suggesting that recruitment of compatible strains from the soil microbiome by the plants occurred. In addition, the structure and composition of diazotroph/N scavenger bacterial communities differed between tomato and lulo, and these plant species showed a contrasting overall response to the inoculation, where several strains resulted in decreases in tomato biomass, while the opposite was observed for lulo. The data in the literature on the identification and selection of PGPB have no consensus with respect to the best strategy to be used in the search for elite bacterial strains, although a common thread reported is the importance of the use *in vivo* trials to select for growth-promoting bacteria [81,96]. In this same sense, autochthonous plant growth-promoting bacterial strains are believed to have relatively high potential in eliciting growth-promoting effects, which is in accordance with the evolutionary and specific needs for successful chemical crosstalk, despite variations in this specificity being vastly reported [95,97,98].

Among the 63 total diazotrophic/N-scavenging strains that promoted increases in tomato and/or lulo biomass (RDW, SDW or both) reported here, those isolated from TR samples were predominant (34 strains), suggesting that *S. lycopersicum* is more efficient than *S. quitoense* at selecting PGPB from the soil microbiome. On the other hand, only a few strains showed a positive inoculation effect on tomato (28 strains, 17 autochthonous ones of tomato), while the biomass of lulo increased in response to the inoculation of a large number of strains (55 strains, 15 autochthonous ones of lulo); these results suggested that lulo is more easily influenced by PGPB than is tomato. Notably, strains ranked by the bonitur scale showing scores greater than 10 (the top 16 strains) caused increases in biomass for one or both *Solanum* species, indicating that this strategy fits well for the selection of candidates for additional inoculation experiments under field conditions. The best plant growth-promoting bacterial candidates included 8 *Rhizobium* species, 4 *Enterobacter* species, 3 *Pseudomonas* species and one *Cupriavidus* species, of which the strains *Pseudomonas* sp. 23S and *Cupriavidus* sp. 26S were effective for both *Solanum* species and may have potential for applications in other horticultural crop species. The high number of *Rhizobium* strains isolated from *Solanum* plants at population densities greater than $1 \times 10^4$ cells $g^{-1}$ plant material raises important questions concerning the ability of representatives from this phylogenetic group to act as opportunistic or specific associative bacteria, which future genomic studies should answer. This study is the first report describing the isolation and characterization of bacterial representatives associated with lulo (*S. quitoense*) and their assessment as PGPB in association with this *Solanum* species. Previous studies have reported on root fungal communities associated with lulo plantations as well as the response of lulo to inoculation with arbuscular mycorrhizal fungi (AMF) [99,100], but to date, no studies have investigated the association of lulo and diazotrophic/N-scavenging bacteria as growth promoters.

## Supporting information

**S1 File. Reconstruction of phylogenetic relationships between the isolated strains described in the present study and the respective type species for each genus identified based on the alignment of 16S rRNA gene sequences.**
(PDF)

**S1 Table. Chemical characteristics of soils used in this study.**
(DOCX)

**S2 Table. Number of diazotroph/N scavenger bacterial strains isolated from soils under different management conditions and from unwashed roots of tomato and lulo plants grown on those soils, according to findings in different semisolid N-free culture media.**
(DOCX)

**S3 Table. Molecular characterization and plant growth-promoting traits of the 101 diazotroph/N scavenger bacterial strains described in this study and the respective ranking of their biotechnological potential to promote the growth of *Solanum*, according to the bonitur scale.**
(DOCX)

**S1 Fig. Plant growth-promoting mechanisms of the 101 diazotroph/N scavenger bacterial strains described in this study.** The means plotted in the same color represent groups that do not significantly differ at $p < 0.05$ according to the Scott-Knott algorithm. The bars refer to the maximum and minimum values for each plot. (A) IAA production, (B) $FePO_4$ solubilization,

(C) AlPO$_4$ solubilization and (D) siderophore production.
(TIFF)

**S2 Fig.** Venn diagram (A) showing unique and shared diazotrophic/N-scavenging bacteria that significantly (Scott-Knott algorithm, $p < 0.05$) increased the root dry weight (RDW) and shoot dry weight (SDW) of tomato and lulo plants grown under N-limiting conditions compared to those of control plants (Fig 4). Correlation analysis (B) of biomass accumulation of the roots and aerial parts of tomato and lulo plants in response to inoculation with 101 different diazotrophic/N-scavenging bacteria.
(TIF)

**S3 Fig. Correlation matrix of associations between the biochemical characterization of strains and the biometric parameters of inoculated plants and their bonitur scale ranking.** IAA, indole-3-acetic acid production; FePO$_4$, FePO$_4$ solubilization; AlPO$_4$, AlPO$_4$ solubilization; Sider, Siderophore production; RDW, root dry weight of either tomato or lulo; SDW, shoot dry weight of either tomato or lulo; Rank pts, ranking according to the bonitur scale.
(TIF)

**S1 Raw Images. Raw gel images used to produce the dendrogram estimated by the cluster analysis of BOX-PCR fingerprints of diazotrophic/N-scavenging bacteria isolated from different sources and from soils under different management conditions.**
(PDF)

## Acknowledgments

The authors acknowledge the INCT—Plant Growth-Promoting Microorganisms for Agricultural Sustainability and Environmental Responsability (INCT-MPCPAgro), the CNPq (conv. No. 458099/2014-7), the Fundação Araucária and the Universidade Estadual de Londrina for the financial support. The authors thank the Coordenação de Aperfeiçoamento de Pessoal de Nível Superior (CAPES) for granting scholarships to Mónica Y. A. Zuluaga and Karina M. L. Milani.

## Author Contributions

**Conceptualization:** Mónica Yorlady Alzate Zuluaga, André Luiz Martinez de Oliveira.

**Data curation:** Mónica Yorlady Alzate Zuluaga, Karina Maria Lima Milani.

**Formal analysis:** Mónica Yorlady Alzate Zuluaga, Leandro Simões Azeredo Gonçalves.

**Funding acquisition:** André Luiz Martinez de Oliveira.

**Investigation:** Mónica Yorlady Alzate Zuluaga, Karina Maria Lima Milani, Leandro Simões Azeredo Gonçalves, André Luiz Martinez de Oliveira.

**Methodology:** Mónica Yorlady Alzate Zuluaga, Karina Maria Lima Milani.

**Resources:** Leandro Simões Azeredo Gonçalves, André Luiz Martinez de Oliveira.

**Software:** Leandro Simões Azeredo Gonçalves.

**Supervision:** Leandro Simões Azeredo Gonçalves, André Luiz Martinez de Oliveira.

**Validation:** Leandro Simões Azeredo Gonçalves.

**Visualization:** Mónica Yorlady Alzate Zuluaga, André Luiz Martinez de Oliveira.

**Writing – original draft:** Mónica Yorlady Alzate Zuluaga.

**Writing – review & editing:** Mónica Yorlady Alzate Zuluaga, Karina Maria Lima Milani, Leandro Simões Azeredo Gonçalves, André Luiz Martinez de Oliveira.

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
