## [Decision Letter · Decision Letter 0]

4 Dec 2019

PONE-D-19-29936

Diversity and plant growth-promoting functions of diazotrophic/N-scavenging bacteria isolated from the soils and rhizospheres of two species of Solanum

PLOS ONE

Dear Prof. Andre Luiz Martinez de Oliveira,

Thank you for submitting your manuscript to PLOS ONE. After careful consideration, we feel that it has merit but does not fully meet PLOS ONE’s publication criteria as it currently stands. Therefore, we invite you to submit a revised version of the manuscript that addresses the points raised during the review process.

There are some missing materials that should be added to the MS. 

Regarding blot/gel data: PLOS ONE now requires that submissions reporting blots or gels include original, uncropped blot/gel image data as a supplement or in a public repository. This is in addition to complying with our image preparation guidelines described at https://journals.plos.org/plosone/s/figures#loc-blot-and-gel-reporting-requirements. These requirements apply both to the main figures and to cropped blot/gel images included in Supporting Information.

Briefly, figures reporting blot or gel images comply with the journalâ€™s image preparation guidelines and that the original data are provided following the journalâ€™s requestIf you need additional information, please pay specific attention to the journal's scope and policies as detailed on the following pages: i) Scope: https://journals.plos.org/plosone/s/journal-information#loc-scope ii) Editorial policies including those in relation to publication ethics, research ethics, and data sharing: https://journals.plos.org/plosone/s/editorial-and-publishing-policies.

In addition, according to the comments of Reviewer #2 the English should be checked by an English native speaker.

We would appreciate receiving your revised manuscript by Jan 18 2020 11:59PM. To enhance the reproducibility of your results, we recommend that if applicable you deposit your laboratory protocols in protocols.io, where a protocol can be assigned its own identifier (DOI) such that it can be cited independently in the future. For instructions see: http://journals.plos.org/plosone/s/submission-guidelines#loc-laboratory-protocols

We look forward to receiving your revised manuscript.

Kind regards,

Luigimaria Borruso

Academic Editor

PLOS ONE

Journal Requirements:

Additional Editor Comments (if provided):

Reviewers' comments:

Reviewer's Responses to Questions

**Comments to the Author**

1. Is the manuscript technically sound, and do the data support the conclusions?

Reviewer #1: Yes

Reviewer #2: Yes

2. Has the statistical analysis been performed appropriately and rigorously? 

Reviewer #1: Yes

Reviewer #2: Yes

3. Have the authors made all data underlying the findings in their manuscript fully available?

Reviewer #1: Yes

Reviewer #2: Yes

4. Is the manuscript presented in an intelligible fashion and written in standard English?

Reviewer #1: Yes

Reviewer #2: No

5. Review Comments to the Author

Reviewer #1: The article "Diversity and plant growth-promoting functions of diazotrophic/N-scavenging bacteria isolated from the soils and rhizospheres of two species of Solanum" is a culturomics study on the rhizosphere of two related plant species in three different agricultural management. On top of it, the authors carried a phenotypic characterization of the 101 isolated strains and genomic fingerprinting. This article has a clear experimental design and the methods are adequate to the questions. The results are well detailed and the discussion is a critical analysis of the results in light of the literature.

In my opinion, this article meets the criteria for a direct acceptance, as I could not find any major or minor revision to be applied.

Reviewer #2: The manuscript by Alzate Zuluaga and co-workers presents the intensive isolation of promising PGPB strains with potential biotechnological features and their characterization for the PGPR traits and their in vivo activities.

the research is well designed and conducted; the results are well presented and discussed in the frame of the existing relevant literature.

In my opinion, the manuscript is suitable to be published in POLOS ONE. However, while reading, I was able to spot several imperfections in the English writing, therefore I suggest the authors to have their manuscript revised by a native English speaker before submitting the final version for publication.

6. PLOS authors have the option to publish the peer review history of their article (what does this mean?). If published, this will include your full peer review and any attached files.

Reviewer #1: No

Reviewer #2: No

---

## [Author Response · Author response to Decision Letter 0]

9 Dec 2019

Response to Reviewer 1

Thank for your kind comments.

Response to Reviewer 2

Thank for your kind comments. In fact, the manuscript had been revised by the American Journal Experts (AJE) team, which certificate is attached for PLOS Journal records. Nevertheless, when preparing the Revised Manuscript version, some missing revisions were found and we expect that these points are now covered and corrected.

---

## [Editor Report · Decision Letter 1]

19 Dec 2019

Diversity and plant growth-promoting functions of diazotrophic/N-scavenging bacteria isolated from the soils and rhizospheres of two species of Solanum

PONE-D-19-29936R1

Dear Prof. Andre Luiz Martinez de Oliveira,

We are pleased to inform you that your manuscript has been judged scientifically suitable for publication and will be formally accepted for publication once it complies with all outstanding technical requirements.

With kind regards,

Luigimaria Borruso

Academic Editor

PLOS ONE
---

## [Editor Report · Acceptance letter]

26 Dec 2019

PONE-D-19-29936R1 

Diversity and plant growth-promoting functions of diazotrophic/N-scavenging bacteria isolated from the soils and rhizospheres of two species of Solanum 

Dear Dr. Martinez de Oliveira:

I am pleased to inform you that your manuscript has been deemed suitable for publication in PLOS ONE. Congratulations! Your manuscript is now with our production department. 

With kind regards,

on behalf of

Dr. Luigimaria Borruso 

Academic Editor

PLOS ONE